# Robust Trajectory Distillation: Hybrid Reweighting Meets Teacher-Inspired Targets

## Abstract

Robust training under noisy labels remains a critical challenge in deep learning due to the risk of confirmation bias and overfitting in iterative correction pipelines. In this work, we propose a novel trajectory-based dataset distillation framework that jointly addresses noise suppression and knowledge preservation without requiring label correction or clean subsets. Our method introduces two complementary components: Selective Guidance Reweighting (SGR) and Teacher-Inspired Auxiliary Targets (TIAT). SGR improves teacher signal quality by integrating global forgetting patterns (via second-split forgetting) with local feature consistency (via KNN-based evaluation), forming a hybrid reweighting mechanism that prioritizes clean supervision. TIAT further enhances the learning capacity by injecting auxiliary guidance derived from intermediate teacher dynamics, ensuring internal consistency while reinforcing informative signals. Together, these strategies enable the distilled dataset to retain cleaner and richer knowledge representations under noisy supervision. The proposed framework is label-preserving, computationally efficient, and broadly applicable. Extensive experiments on benchmark datasets demonstrate consistent performance improvements over state-of-the-art dataset distillation methods across symmetric, asymmetric, and real-world noise scenarios.

## 1 Introduction

The growing complexity of deep learning models has heightened the dependence on large-scale, high-quality datasets. However, real-world annotations are frequently affected by label noise, stemming from factors such as human bias, inconsistencies in crowdsourced labeling, semantic ambiguities, and errors in web-crawled pseudo-labels. Although manual relabeling could, in principle, address this issue, it is often infeasible due to the substantial cost associated with correcting datasets containing millions of samples (e.g. WebVision (Li et al., 2017)). As a result, developing robust learning methods that can effectively handle noisy labels has emerged as a critical research challenge.

To mitigate the impact of noisy labels, a range of strategies has been proposed, including sample selection (Malach & Shalev-Shwartz, 2017; Liu et al., 2020; Zhu et al., 2022), loss weighting (Zhang & Sabuncu, 2018; Liu & Guo, 2020), and label correction (Reed et al., 2014; Song et al., 2019). These approaches typically aim to estimate the likelihood of label corruption and adapt the training process accordingly, either by down-weighting or excluding potentially noisy samples, or by explicitly correcting their labels. While such online, iterative optimization schemes have demonstrated effectiveness in controlled experimental settings, the lack of reliable ground-truth validation anchors (e.g., a clean subset) complicates the stabilization of the optimization trajectory, increasing the risk of overfitting to unvalidated pseudo-labels and triggering self-reinforcing cycles of error amplification.

The limitations of conventional noisy label learning methods primarily arise from the tight coupling between noise estimation and model training. At the core of these approaches is a self-referential dilemma: the model simultaneously serves as both noise detector and target predictor, creating a closed-loop system prone to confirmation bias amplification. This feedback loop often results in overconfidence in incorrect labels, ultimately degrading generalization performance. Furthermore, such methods typically depend on stable external anchors (e.g., verified clean subsets or other prior knowledge) and resource-intensive iterative procedures, which increase computational costs and deployment complexity. In real-world applications, these constraints significantly impede scalability. Additionally, dynamic relabeling introduces security risks, as adaptive data modifications

unintentionally expose privacy-sensitive information. Recent advances (Cheng et al., 2024) suggest dataset distillation as a promising alternative: by synthesizing compact training subsets that preserve essential information, it enhances robustness under noisy supervision.

Although previous work (Cheng et al., 2024) has demonstrated the potential of dataset distillation in mitigating label noise, significant limitations remain regarding the efficiency and capacity of noise-to-clean knowledge transfer. Conventional distillation frameworks (Wang et al., 2018b; Cui et al., 2022) suffer from two key shortcomings: **(1) Noise-Agnostic Information Extraction:** Methods such as DANCE (Zhang et al., 2024a) and DATM (Guo et al., 2024) generally assume clean supervision and lack explicit mechanisms to suppress feature associations induced by corrupted labels. Consequently, their performance degrades under noise. For example, in symmetric noise with $20\%$ noise rate on the CIFAR-10 dataset, both DATM and DANCE exhibit an accuracy drop of approximately $2\%$ to $3\%$ compared to their performance in clean environments when the images per class (IPC) is 50. **(2) Capacity-Constrained Synthesis:** Synthetic datasets are typically parameterized as fixed-size image tensors (e.g., 50 IPC), which imposes a strict upper bound on their capacity to represent informative content. This limitation increases the risk of premature information compression, potentially discarding useful clean signals before sufficient disentanglement from noisy patterns can occur. These limitations motivate a central question: *How can we improve both the quality and capacity of distilled data under noisy supervision?* This challenge can be broken down into two complementary objectives:

**Challenge ❶:** Improving the fidelity of the synthetic dataset by promoting cleaner knowledge retention and enhancing robustness to label noise through better teacher signal quality.

**Challenge ❷:** Complementing intrinsic knowledge quality improvements by enhancing the effectiveness of knowledge transmission from teacher to student trajectories, allowing the synthetic dataset to absorb richer and cleaner supervision even under noise.

To this end, we propose two complementary techniques: **Selective Guidance Reweighting (SGR)** and **Teacher-Inspired Auxiliary Targets (TIAT)**, respectively. SGR enhances synthetic data learning by improving teacher trajectory quality. It integrates global forgetting trends via second-split forgetting (Maini et al., 2022; Li et al., 2022) with local neighborhood consistency through KNN-based evaluation, forming a hybrid reweighting scheme. This dynamic calibration ensures that the teacher's guidance focuses more on clean supervision, thereby improving the overall teaching quality and enabling the student to learn more effectively. The underlying assumption is straightforward: a more reliable teacher produces a more capable student. TIAT are designed to further augment the distillation process by providing additional high-quality supervision signals beyond the primary teacher trajectory. Specifically, TIAT derives a set of auxiliary targets—akin to residual teaching signals—from the teacher's own training dynamics, such as intermediate predictions or consistency-based feedback. These auxiliary targets act as complementary assignments that reinforce the primary supervision, helping the student model to consolidate and generalize the distilled knowledge more effectively. Importantly, although these signals are decoupled from the trajectory itself, they are grounded in the teacher's behavior and thus maintain internal consistency within the distillation framework. By injecting such coherent, yet enriched guidance, TIAT enables the synthetic dataset to absorb more informative supervision without introducing conflicting objectives.

In summary, we propose a distillation framework specifically tailored for learning under noisy supervision. Conceptually, the framework resembles a teacher who not only refines their expertise to provide higher-quality instruction but also assigns "homework-like" auxiliary tasks that further reinforce and consolidate the student's learning outcomes. Our contributions are as follows:

- We introduce a progressive mechanism that integrates both dynamic and static sample reweighting, enabling the fusion of diverse teacher trajectory signals to achieve robust and superior noise suppression.
- We propose an auxiliary guidance regularization strategy that ensures clean trajectory consistency during distillation, effectively strengthening the influence of clean samples throughout the process.
- Our framework is label-preserving and computationally efficient, requiring no relabeling or extensive retraining, making it practical for real-world noisy data scenarios.

- Extensive experiments demonstrate that our framework consistently improves distillation performance across diverse datasets and noise settings, validating its generality and robustness.

## 2 RELATED WORKS

### 2.1 LEARNING WITH NOISY LABELS

Real-world datasets often contain corrupted labels due to annotation errors or automated collection. To address this, noisy label learning has developed three major strategies: *sample selection*, *label correction*, and *sample reweighting*. Sample selection methods aim to identify clean data during training, typically using loss-based filtering. A representative work is Decoupling (Malach & Shalev-Shwartz, 2017), which introduced the small-loss trick. Follow-up works (Jiang et al., 2018; Han et al., 2018) incorporate external guidance or co-training, while curriculum-based methods (Lyu & Tsang, 2019; Zhou et al., 2021) and early-learning regularization (Liu et al., 2020) enhance robustness via dynamic filtering or temporal consistency. Some approaches even detect noisy samples prior to training (Zhu et al., 2022; Wang et al., 2018a). Label correction methods refine labels using model predictions (Reed et al., 2014; Zhou et al., 2024), semi-supervised strategies (Li et al., 2020), or meta-learning and neighborhood consistency (Tu et al., 2023; Li et al., 2022). Sample reweighting strategies (Zhang & Sabuncu, 2018; Shu et al., 2019; Di Salvo et al., 2024) instead adjust loss contributions based on sample reliability. Additionally, K-NN-based methods (Bahri et al., 2020; Iscen et al., 2022) have proven effective for noise detection by evaluating local label consistency in the feature space. Building on this, dataset distillation has recently emerged as a promising alternative. As shown in (Cheng et al., 2024), it addresses limitations such as iterative error amplification and privacy concerns, while offering strong performance under noisy supervision.

### 2.2 DATASET DISTILLATION

Dataset distillation (Sachdeva & McAuley, 2023; Wang et al., 2018b; Cui et al., 2022) aims to compress a large dataset into a compact synthetic set while preserving downstream performance. The foundational work (Wang et al., 2018b) introduced the idea of optimizing synthetic data to match training dynamics observed on real data. Subsequent research has evolved along three major paradigms: *meta-learning*, *parameter matching*, and *distribution matching*. Meta-learning methods (Nguyen et al., 2021; Zhou et al., 2022; Loo et al., 2023) adopt a bi-level optimization framework to generalize across model initializations but suffer from high computational overhead. Parameter matching, including gradient (Zhao et al., 2021) and trajectory matching (Cazenavette et al., 2022; Du et al., 2023), directly aligns model updates between real and synthetic data, with trajectory-based methods achieving state-of-the-art results (Guo et al., 2024). Variants further explore progressive optimization (Chen et al., 2023), hybrid data composition (Lee & Chung, 2024), and group-wise structures (He et al., 2024). Distribution matching offers an efficient alternative by aligning real and synthetic data distributions in feature space (Zhao & Bilen, 2022; Zhang et al., 2024b), class relations (Deng et al., 2024), or image-label correlations (Zhang et al., 2024a), without requiring nested optimization. Recent advances even reformulate this as neural feature alignment (Wang et al., 2025). However, most methods assume clean supervision and degrade under real-world label noise. To address this limitation, we propose a noise-aware extension of trajectory matching that explicitly models and suppresses corruption during both the distillation and deployment phases.

## 3 PRELIMINARIES

### 3.1 NOISY DATASET DISTILLATION

Let $\mathcal{D}_{\text{real}} = \{(x_i, \tilde{y}_i)\}_{i=1}^N$ denote a real-world training dataset, where $\tilde{y}_i$ is the observed (and potentially noisy) label for input $x_i$. We assume the existence of label noise such that $\tilde{y}_i \neq y_i^*$ for some $i$, where $y_i^*$ is the clean but unobserved ground-truth label. In contrast, the test set $\mathcal{D}_{\text{test}}$ is assumed to be entirely clean and representative of the true data distribution, and is used to evaluate generalization performance. Our objective is to synthesize a compact dataset $\mathcal{S}$, with $|\mathcal{S}| \ll |\mathcal{D}_{\text{real}}|$, such that a model trained solely on $\mathcal{S}$ achieves lower generalization error on $\mathcal{D}_{\text{test}}$ than one trained on the full

noisy dataset $\mathcal{D}_{\text{real}}$. Formally, the distillation task can be formulated as the following optimization problem:

$$\mathcal{S}^* = \arg\min_{\mathcal{S}} \mathcal{L}(\mathcal{S}, \mathcal{D}_{\text{real}}), \tag{1}$$

where $\mathcal{L}$ is a general objective function. In our setting, it is instantiated as a trajectory matching loss (Guo et al., 2024) that aligns the student's learning dynamics (trained on $\mathcal{S}$) with those of a teacher model trained on $\mathcal{D}_{\text{real}}$.

Trajectory matching-based dataset distillation methods typically adopt a bilevel optimization scheme comprising an *inner loop* and an *outer loop*. The inner loop simulates the training dynamics of a student model on the current synthetic dataset $\mathcal{S}$, while the outer loop updates $\mathcal{S}$ such that the student's optimization trajectory closely matches that of a teacher trained on the real dataset $\mathcal{D}_{\text{real}}$.

Specifically, a teacher model is first trained on $\mathcal{D}_{\text{real}}$ to produce a reference trajectory $\tau^* = \{\theta_t^*\}_{t=1}^T$, where $\theta_t^*$ denotes the model parameters at iteration $t$. Meanwhile, the synthetic dataset $\mathcal{S}$ is initialized—either by sampling from Gaussian noise or selecting real samples—with corresponding soft or hard labels.

In the *inner loop*, we simulate student learning dynamics by iteratively updating parameters using $\mathcal{S}$. Given a starting point $\hat{\theta}_t = \theta_t^*$, the student parameters are updated over $N$ steps according to:

$$\hat{\theta}_{t+n+1} = \hat{\theta}_{t+n} - \alpha \nabla_{\hat{\theta}_{t+n}} \ell(\mathcal{A}(\mathcal{S}); \hat{\theta}_{t+n}), \tag{2}$$

where $\ell$ denotes the task loss (e.g., cross-entropy), $\mathcal{A}(\mathcal{S})$ denotes a mini-batch drawn from $\mathcal{S}$ with optional data augmentation, and $\alpha$ is the learning rate. This simulates the student trajectory $\hat{\tau} = \{\hat{\theta}_{t+n}\}_{n=1}^N$ induced by $\mathcal{S}$.

In the *outer loop*, we optimize the synthetic dataset $\mathcal{S}$ such that the student's final parameters align with the teacher's future trajectory. Let $\mathcal{T} = \{t_1, t_2, \ldots, t_K\}$ be a set of anchor steps. At each $t \in \mathcal{T}$, the teacher parameters after $M$ additional steps are denoted as $\theta_{t+M}^*$. In our method, $\mathcal{L}$ in Eq. 1 is instantiated as the following normalized trajectory alignment loss:

$$\mathcal{L}_{\text{traj}}(\mathcal{S}, \mathcal{D}_{\text{real}}) = \frac{\left\| \hat{\theta}_{t+N} - \theta_{t+M}^* \right\|_2^2}{\left\| \theta_t^* - \theta_{t+M}^* \right\|_2^2}. \tag{3}$$

The normalization term $\left\| \theta_t^* - \theta_{t+M}^* \right\|_2^2$ calibrates the alignment error relative to the teacher's own parameter change, making the loss scale-invariant across steps. The numerator measures how closely the student, trained on $\mathcal{S}$, approximates the teacher's future parameters, while the denominator normalizes for the teacher's update magnitude to ensure scale invariance. This outer-loop objective is used to update $\mathcal{S}$ via gradient-based optimization, enabling the synthetic data to induce faithful learning dynamics that reflect those observed on real data.

# 4 METHODOLOGY

## 4.1 OVERVIEW

Figure 1 illustrates the overall workflow of our proposed framework for noisy dataset distillation, which aims to construct a compact synthetic dataset $\mathcal{S}$ from a noisy real-world dataset $\mathcal{D}_{\text{real}} = (x_i, \tilde{y}_i)$. The core idea is to enable the student model trained on $\mathcal{S}$ to replicate the learning dynamics of a teacher model trained on $\mathcal{D}_{\text{real}}$, while suppressing the adverse effects of label noise. To this end, our framework follows a three-stage pipeline. First, we train a teacher model on $\mathcal{D}_{\text{real}}$ to obtain a reference parameter trajectory $\tau^* = \theta_t^*$; during this stage, we apply *Selective Guidance Reweighting (SGR)* (See Sec. 4.2) to assign sample-specific weights based on reliability estimates derived from second-split forgetting and KNN-based local consistency, thereby producing a cleaner supervision signal. Second, we initialize a synthetic dataset $\mathcal{S}$ with soft or hard labels and simulate the inner-loop training dynamics of a student model on $\mathcal{S}$. In this stage, we further incorporate *Teacher-Inspired Auxiliary Targets (TIAT)* (See Sec. 4.3) —a set of auxiliary consistency signals extracted from intermediate teacher states over high-confidence samples — to boost supervision beyond trajectory alignment. Finally, in the outer loop, we optimize $\mathcal{S}$ via a normalized trajectory matching loss computed across

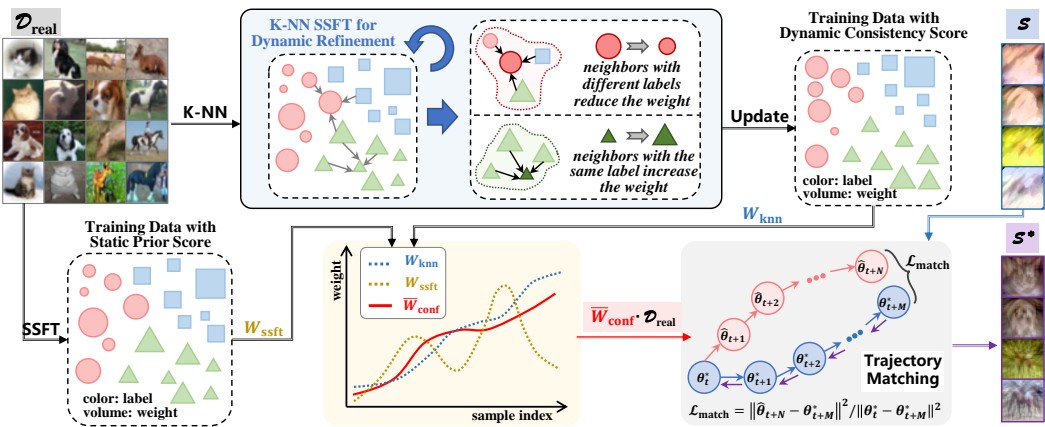

Figure 1: **The overall pipeline.** Our proposed pipeline includes two main contributions: (1) during teacher trajectory training, we apply sample-specific weighting adjustments to modulate the influence of each sample based on its estimated reliability; and (2) in the subsequent distillation phase, we utilize a subset of high-confidence samples to impose additional constraints and regularization, thereby enhancing the quality and generalization of the distilled dataset.

multiple anchor steps, aligning the student's learned trajectory with that of the teacher. This bilevel optimization process jointly improves the quality of supervision (via SGR) and the effectiveness of trajectory alignment (via TIAT), enabling the distilled dataset to retain cleaner, more generalizable knowledge even under noisy conditions.

## 4.2 SELECTIVE GUIDANCE REWEIGHTING

We propose **Selective Guidance Reweighting (SGR)** to control the fidelity of signals used for synthetic data optimization. SGR introduces a hybrid reliability estimator for each training sample $i$, composed of:

- a **dynamic consistency score** $W_{\text{knn}}^{(i)}$ based on feature-space neighborhood agreement.
- a **static prior score** $W_{\text{ssft}}^{(i)}$ derived from sample forgetting behavior;

These scores are combined into a unified weight $W^{(i)}$ via time-dependent convex interpolation.

**Dynamic Estimation via KNN.** Given the predicted label distributions $\hat{y}_j$ of the $K$ nearest neighbors of the observed sample $i$ in the feature space, we define its KNN consistency score as:

$$W_{\text{knn}}^{(i)} = 1 - \frac{1}{K} \sum_{j=1}^{K} \text{JS}(\bar{y}_i, \hat{y}_j), \tag{4}$$

where $\text{JS}(\cdot, \cdot)$ denotes Jensen-Shannon divergence. This score ranges in $[0, 1]$, with higher values indicating better local agreement.

**Static Prior via SSFT.** Inspired by (Maini et al., 2022), we introduce a forgetting-based difficulty score to quantify the reliability of each training sample. For each sample $i$, we record:

- $t_{\text{learn}}^{(i)}$: the earliest epoch when the sample is first correctly classified;
- $t_{\text{forget}}^{(i)}$: the earliest epoch after which it is misclassified again.

These timestamps reflect the memorization and retention behavior of a sample during training. To assess sample-level difficulty, we define the SSFT score as:

$$s^{(i)} = \lambda \cdot \frac{t_{\text{learn}}^{(i)}}{\max_j t_{\text{learn}}^{(j)}} + (1 - \lambda) \cdot \left( 1 - \frac{t_{\text{forget}}^{(i)}}{\max_j t_{\text{forget}}^{(j)}} \right), \tag{5}$$

where $\lambda \in [0, 1]$ balances the contributions of learnability and forgettability. A high $s^{(i)}$ implies that sample $i$ was learned late and forgotten early, thus more likely to be noisy or difficult. We convert this difficulty score into a static reliability weight:

$$W_{\text{ssft}}^{(i)} = 1 - s^{(i)}, \tag{6}$$

such that clean and stable samples are assigned higher weights at the beginning of training. This reflects the global memorization difficulty of sample $i$, with higher scores favoring clean samples with high probabilities.

**Hybrid Weighting via Curriculum Fusion.** To balance global priors and evolving local consistency, we define a convex combination:

$$W_t^{(i)} = (1 - \alpha_t) \cdot W_{\text{ssft}}^{(i)} + \alpha_t \cdot W_{\text{knn}}^{(i)}, \tag{7}$$

where $\alpha_t \in [0, 1]$ is a time-dependent blending coefficient that increases linearly over training epochs. Specifically, we set:

$$\alpha_t = \min \left( \frac{t}{T_{\text{warmup}}} \cdot \alpha_{\max}, \alpha_{\max} \right), \tag{8}$$

where $T_{\text{warmup}}$ is a predefined transition period (e.g., 20% of training). Optionally, when training multiple teacher trajectories indexed by $p \in \{1, \dots, P\}$, we set $\alpha_{\max}^{(p)} = \frac{p-1}{P}$ to diversify the static-dynamic balance across teachers (more details can be refer to Eq. 9).

**Remark.** The timestep weight $W_t^{(i)}$ is used to modulate the contribution of sample $i$ when updating the teacher model parameters or computing guidance for $\mathcal{S}$ in outer-loop optimization. This strategy enables the distillation process to prioritize clean, informative signals throughout training.

## 4.3 TEACHER-INSPIRED AUXILIARY TARGETS (TIAT)

While the **Selective Guidance Reweighting (SGR)** module focuses on suppressing noisy signals during teacher trajectory training, it does not explicitly constrain the distilled dataset $\mathcal{S}$ to remain aligned with clean supervision. To address this limitation, we introduce **Teacher-Inspired Auxiliary Targets (TIAT)**, which injects clean-aware regularization into the distillation process by mining a reliable subset and incorporating trajectory-level uncertainty.

**Probabilistic Confidence Aggregation.** Let $W_p^{(i)}$ denote the final-stage reliability weight assigned to sample $i$ by the $p$-th teacher trajectory, trained with static-dynamic blending coefficient $\alpha_{\max}^{(p)}$. To estimate the sample's overall confidence, we define:

$$\bar{W}_{\text{conf}}^{(i)} := \mathbb{E}_{p \sim \mathcal{U}(\mathcal{P})} \left[ W_p^{(i)} \right] \approx \frac{1}{P} \sum_{p=1}^{P} W_p^{(i)}, \tag{9}$$

where $\mathcal{P} = \{1, \dots, P\}$ is the index set of all trajectories. This aggregated score summarizes the expected reliability of sample $i$ across all teacher views.

We then compute the probabilistic trajectory alignment loss $\mathcal{L}_{\text{match}}$ that encourages student updates to follow the aggregated teacher behavior:

$$\mathcal{L}_{\text{match}} = \frac{1}{|\mathcal{S}|} \sum_{i \in \mathcal{S}} \frac{\|\hat{\theta}_{t+N}^{(i)} - \theta_{t+M}^*\|_2^2}{\|\theta_{t+M}^* - \theta_t^*\|_2^2}, \tag{10}$$

where $\hat{\theta}_{t+N}^{(i)}$ denotes the student parameters after training on synthetic sample $i$.

**Uncertainty-Aware Auxiliary Regularization.** To quantify confidence variance across trajectories, we compute:

$$\mathrm{Var}_p(W_p^{(i)}) = \mathbb{E}_{p \sim \mathcal{U}(\mathcal{P})} \left[ \left( W_p^{(i)} - \bar{W}_{\mathrm{conf}}^{(i)} \right)^2 \right]. \tag{11}$$

This variance reflects the stability of confidence scores assigned to sample $i$; low-variance samples are deemed more consistently reliable.

We then define an approximately reliable subset $\mathcal{D}_{\mathrm{sub}}$ using two complementary criteria: $W_{\mathrm{ssft}}^{(i)} \geq \delta_{\mathrm{sup}}$ and $\mathrm{Var}_p(W_p^{(i)}) \leq \sigma_{\mathrm{inf}}$, where $\delta_{\mathrm{sup}}$ and $\sigma_{\mathrm{inf}}$ are fixed thresholds. This set captures samples that are both statistically confident and dynamically stable, forming the basis of our auxiliary regularization. Based on $\mathcal{D}_{\mathrm{sub}}$, we fine-tune $\theta_t^*$ to produce a cleaner teacher checkpoint $\theta_{t+M}^{\mathrm{ft}}$ and define the auxiliary loss as:

$$\mathcal{L}_{\mathrm{aux}} = \frac{1}{|\mathcal{S}|} \sum_{i \in \mathcal{S}} \frac{\|\hat{\theta}_{t+N}^{(i)} - \theta_{t+M}^{\mathrm{ft}}\|_2^2}{\|\theta_{t+M}^{\mathrm{ft}} - \theta_t^*\|_2^2}. \tag{12}$$

We integrate the original and fine-tuned guidance using a convex combination:

$$\mathcal{L}_{\mathrm{total}} = (1 - \beta) \cdot \mathcal{L}_{\mathrm{match}} + \beta \cdot \mathcal{L}_{\mathrm{aux}}, \quad \beta \in [0, 1], \tag{13}$$

where $\beta$ balances the influence of the original and clean-adjusted trajectories.

**Discussion.** TIAT introduces an uncertainty-aware auxiliary supervision signal into dataset distillation, driven by both trajectory consensus and variance-aware selection. By formulating confidence as a probabilistic mean and incorporating uncertainty, the method enables soft guidance that avoids hard filtering while remaining computationally efficient. This module complements SGR by enforcing trajectory-level regularization from the student side, forming a closed-loop distillation pipeline robust to noisy supervision.

## 5 Experiments

### 5.1 Experiment Setup

We evaluate our method on two widely used benchmarks in Noisy Label Learning (LNL) and Dataset Distillation (DD): CIFAR-10, CIFAR-100 (Krizhevsky, 2009). Comprehensive experiments are conducted under varying noise conditions and Image Per Class (IPC) configurations to assess the robustness and scalability of our approach.

**Noisy Settings** In alignment with previous work (Song et al., 2022; Englesson & Azizpour, 2024; Iscen et al., 2022), our evaluation includes both synthetic and real-world noise patterns:

- **Symmetric Noise**: Labels are uniformly corrupted across all non-target classes. Formally, given a noise rate $\eta$, a sample from class $c$ has a probability $\eta$ of being mislabeled as any class $c \neq c'$ with equal likelihood: $P(y_{noisy} = c'|y_{clean} = c) = \frac{\eta}{C-1}, \forall c' \neq c$, where $c$ is the number of classes. We set $\eta \in \{20\%, 40\%\}$.
- **Asymmetric Noise**: Label flips follow class-dependent transition rules (e.g., cat $\leftrightarrow$ dog, truck $\rightarrow$ automobile).
- **Real-World Noise**: We adopt CIFAR-N Wei et al. (2021), a human-annotated variant of CIFAR-10/100 with natural labeling inconsistencies. Its multi-annotator design captures ambiguity patterns, providing a benchmark that aligns with real-world human cognition and a valid noise benchmark.

**Implementation** We compare our method against two baselines: (1) the noisy baseline, which involves training directly on the corrupted dataset, and (2) state-of-the-art distillation approaches, including DATM (Guo et al., 2024), DANCE (Zhang et al., 2024a), and RCIG (Loo et al., 2023). To ensure a fair comparison, we follow the key experimental settings in prior work, particularly DATM. Specifically, we use a three-layer ConvNet for CIFAR-10/100. For evaluation, we report the final mean and standard deviation of test accuracy by training 5 randomly initialized networks with $\mathcal{S}$.

## 5.2 Benchmarking Dataset Distillation Results on Noisy Dataset

As shown in Figure 2, across all four noise scenarios, our method (red curves) has a higher accuracy than the three state-of-the-art baselines—RCIG (purple), DANCE (green), and DATM (blue). Across all settings, our method (red curves) consistently outperforms the baselines, particularly in high IPC regimes. The performance gap is most evident under higher noise (40%), where competing methods degrade rapidly, while our method maintains strong accuracy and smooth trends. Even under the challenging asymmetric noise setting, where existing dataset distillation methods typically suffer substantial performance drops, our approach consistently outperforms the baseline at both 20% and 40% noise levels. These results further confirm that our clean-aware sample reweighting and trajectory-informed distillation improve robustness against label corruption, especially when training data is both scarce and noisy. Table 1 compares our method with recent state-of-the-art approaches on **CIFAR-100N** and **CIFAR-100** under symmetric and asymmetric noise at 20% and 40% ratios. Across all IPC levels, our method achieves the best performance in nearly all settings. Notably, under 40% symmetric noise, we outperform prior methods by large margins—achieving **52.8%** at 100 IPC and **43.2%** at 10 IPC. Even in the challenging "**worse**" case, our method yields the highest result (**45.9%** at 50 IPC). These results highlight the robustness of our clean-aware reweighting and distillation design under severe label noise.

Table 1: Test accuracy (%) of our method and existing state-of-the-art methods on the **CIFAR-100N** and **CIFAR-100** with (a)symmetric noise ratios of 20% and 40%.

| Noise Type | Symmetric | | | | | | Asymmetric | | | | | | worse | |
|---|---|---|---|---|---|---|---|---|---|---|---|---|---|---|
| Noise Ratio | 20% | | | 40% | | | 20% | | | 40% | | | 40.20% | |
| IPC | 10 | 50 | 100 | 10 | 50 | 100 | 10 | 50 | 100 | 10 | 50 | 100 | 10 | 50 |
| **Full Dataset** | 48.9±0.4 | | | 39.9±0.2 | | | 46.2±0.5 | | | 33.0±0.1 | | | 44.4±0.3 | |
| **RCIG** (Loo et al., 2023) | 41.2±0.3 | 38.4±0.3 | - | 36.5±0.4 | 30.7±0.2 | - | 38.9±0.4 | 37.5±0.2 | - | 28.7±0.4 | 27.9±0.3 | - | 37.0±0.3 | 35.3±0.2 |
| DANCE (Zhang et al., 2024a) | 45.8±0.2 | 48.0±0.3 | 47.5±0.2 | 39.7±0.2 | 42.0±0.4 | 42.6±0.3 | 43.8±0.3 | 46.7±0.3 | 48.2±0.4 | 32.2±0.4 | 34.2±0.2 | 35.22±0.3 | **42.0±0.3** | 43.6±0.3 |
| DATM (Guo et al., 2024) | 45.1±0.1 | 49.7±0.3 | 48.9±0.2 | 40.6±0.4 | 45.1±0.4 | 44.4±0.3 | 40.3±0.4 | 45.4±0.3 | 50.2±0.3 | 29.4±0.4 | 32.0±0.3 | 36.0±0.3 | 39.9±0.5 | 43.9±0.2 |
| **Ours** | **45.8±0.3** | **51.6±0.1** | **55.7±0.2** | **43.2±0.2** | **48.4±0.4** | **52.8±0.2** | **44.6±0.2** | **50.3±0.3** | **54.4±0.1** | **34.7±0.1** | **36.5±0.4** | **40.7±0.4** | 41.0±0.3 | **45.9±0.1** |

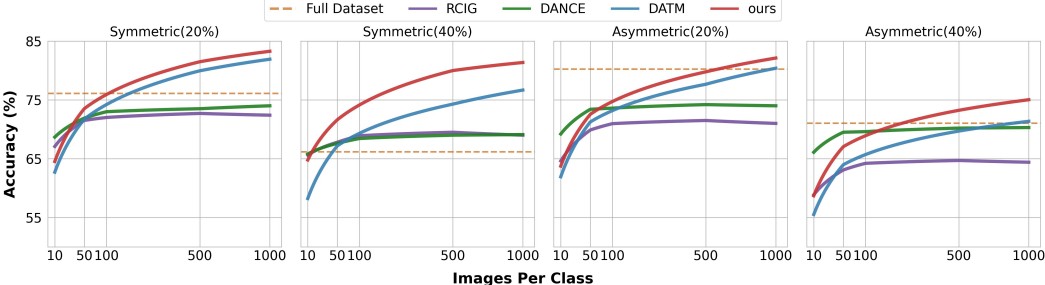

Figure 2: Test accuracy (%) of our method and existing state-of-the-art methods on the **CIFAR-10** with (a)symmetric noise ratios of 20% and 40%.

## 5.3 Ablation Studies

**The Impact of Each Component.** We conduct an ablation study on the CIFAR-10 dataset under symmetric label noise levels of 20% and 40%, evaluating model performance across varying data condensation settings (IPC ∈ 10, 50, 500, 1000). Starting from the DATM baseline, we progressively incorporate our proposed components: *Selective Guidance Reweighting* (SGR) and *Teacher-Inspired Auxiliary Targets* (TIAT). As shown in Table 2, introducing SGR yields consistent performance gains, particularly under severe noise and limited data (e.g., +4.7% at 10 IPC with 40% noise), demonstrating the effectiveness of trajectory-level denoising. Adding TIAT further improves accuracy, with additional gains such as +5.7% at 500 IPC under 40% noise, highlighting the benefit of leveraging uncertainty-filtered clean supervision. Overall, the combination of SGR and TIAT leads to stable and substantial improvements, validating the robustness and compatibility of our framework in noisy learning environments.

**Effectiveness of Diverse Sampling** To assess the robustness of progressive trajectory diversity, we compare *Diverse Sampling*—where $P$ teacher trajectories are trained with uniformly distributed $\alpha_{\max}$

Table 2: Ablation results on CIFAR-10 under symmetric noise with 20% and 40% corruption ratios, evaluated across different IPCs (images per class).

| IPC Method | Symmetric (20%) | | | | Symmetric (40%) | | | |
|---|---|---|---|---|---|---|---|---|
| | 10 | 50 | 500 | 1000 | 10 | 50 | 500 | 1000 |
| DATM | 62.7 | 71.7 | 80.0 | 81.9 | 58.2 | 67.2 | 74.3 | 76.7 |
| + SGR | 63.8 (↑1.1) | 72.8 (↑1.1) | 81.2 (↑1.2) | 83.3 (↑1.4) | 62.9 (↑4.7) | 71.1 (↑3.9) | 79.4 (↑5.1) | 81.2 (↑4.5) |
| + SGR + TIAT | **64.5** (↑1.8) | **73.5** (↑1.8) | **81.5** (↑1.5) | **83.3** (↑1.4) | **64.8** (↑6.6) | **71.6** (↑4.4) | **80.0** (↑5.7) | **81.4** (↑4.7) |

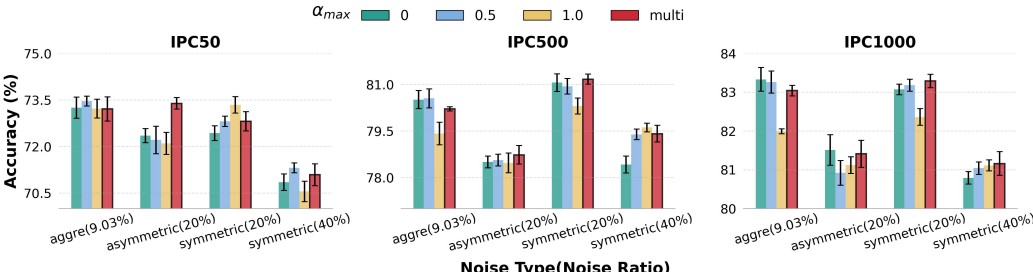

Figure 3: Performance comparison between *Diverse Sampling* and *Fixed Sampling* under various settings on **CIFAR-10**. **Baseline** refers to DATM without our method. **Diverse Sampling** and **Fixed Sampling** incorporate our **Selective Guidance Reweighting** into DATM, where Fixed Sampling uses fixed $\alpha_{max}$ values (**0, 0.5, 1.0**) for teacher trajectory training, while Diverse Sampling **multi** follows the strategy outlined in Section4.2.

values in $[0, 1]$—against a *Fixed Sampling* baseline with constant $\alpha_{max}$. As shown in Figure 3, under increasing noise levels, the Diverse Sampling strategy consistently outperforms fixed configurations, exhibiting stable performance without noticeable degradation. In contrast, fixed-$\alpha$ trajectories show varying sensitivity to the noise rate, indicating their lack of generalizability. These results highlight that diversity in teacher signal strength improves resilience to label corruption, enabling more robust and adaptive supervision during distillation.

Table 3: Ablation study on the effect of $\beta$ under symmetric noise (20% and 40%) on CIFAR-10. We highlight **the top two accuracies** and  the default $\beta$ .

| Subset Ratio $\beta$ | Symmetric (20%) | | | Symmetric (40%) | | |
|---|---|---|---|---|---|---|
| | 50% | 60% | 70% | 50% | 60% | 70% |
| 0.1 | 73.3±0.11 | **73.5±0.38** | 73.3±0.4 | 71.9±0.2 | 71.5±0.4 | **71.8±0.1** |
| 0.5 | 72.5±0.29 | 73.4±0.12 | **74.0±0.3** | 71.7±0.3 | **72.2±0.3** | 71.4±0.2 |
| 1.0 | 67.4±0.12 | 69.6±0.17 | 71.6±0.3 | 70.0±0.3 | 71.7±0.2 | 71.5±0.1 |

**Effect of High-Confidence Subset Proportion and $\beta$ Coefficient.** We investigate how the clean subset ratio and the auxiliary loss weight $\beta$ affect distillation performance under 20% and 40% symmetric noise on CIFAR-10 (IPC=50). As shown in Table 3, the method is robust across a range of subset ratios (50%–70%) and $\beta$ values. While $\beta = 0.5$ yields the highest accuracy in some settings, $\beta = 0.1$ consistently provides stable performance across noise levels, particularly when combined with a 60% subset. In contrast, $\beta = 1.0$ tends to degrade performance due to overemphasis on auxiliary signals. Based on these results, we adopt $\beta = 0.1$ and a 60% subset as default settings for all main experiments.

## 6 CONCLUSION

We investigate the underexplored problem of dataset distillation under noisy-label settings and identify two key challenges: overfitting to label noise and limited capacity of the synthetic set to retain clean signals. To address these, we introduce Selective Guidance Reweighting and Teacher-Inspired Auxiliary Targets. Experiments on benchmark datasets validate the robustness and effectiveness of our approach, paving the way for future research in noise-resilient dataset distillation.

ETHICS STATEMENT

This work does not involve human subjects, animal experiments, or sensitive personal data. The datasets used (e.g., CIFAR-10/100 and their human-annotated variants CIFAR-10N/100N) are publicly available benchmarks commonly used in noisy-label learning research and do not contain personally identifiable information. Our method focuses on improving the robustness of dataset distillation under label noise through trajectory-based reweighting and auxiliary supervision, without introducing harmful, deceptive, or privacy-invasive applications. We have carefully reviewed the ICLR Code of Ethics and confirm that this submission complies with its principles regarding fairness, transparency, and research integrity. The authors declare no conflicts of interest.

REPRODUCIBILITY STATEMENT

To ensure reproducibility, we provide the following: (1) Full implementation details—including network architecture (3-layer ConvNet), distillation pipeline, SGR/TIAT hyperparameters (e.g., $\alpha$ schedule, $\beta$=0.1, 60% high-confidence subset), and noise protocols—are described in Sections 4 and 5.1, as well as in Table 3. (2) All experiments are averaged over 5 random seeds, with mean and standard deviation reported. (3) While code is not included due to double-blind review, we commit to releasing anonymized implementation upon acceptance to facilitate replication.

LLM USAGE STATEMENT

Large Language Models (LLMs) were used in this work solely as a general-purpose writing assistance tool—for example, to improve grammar, clarify phrasing, or check technical terminology in the manuscript. LLMs did not contribute to the conception of the research idea (e.g., SGR or TIAT design), theoretical analysis, experimental setup, or interpretation of results. All scientific content, including algorithm design, loss formulations, and empirical claims, was developed and verified by the authors. No LLM was used to generate novel technical content or to draft substantial portions of the paper. As required by ICLR policy, we confirm that LLMs are not listed as authors, and we take full responsibility for all content under our names.

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
