# OpenReview forum: "Robust Trajectory Distillation: Hybrid Reweighting Meets Teacher-Inspired Targets"
_ICLR.cc/2026/Conference — ICLR 2026 Conference Withdrawn Submission_

### Official Review · Reviewer_Z8Ps · 2025-10-19

**Soundness:** 1
**Presentation:** 1
**Contribution:** 2
**Rating:** 4
**Confidence:** 4

**Summary:**

This paper tackles the dataset distillation problem in the context of condensing a dataset with noisy labels. The authors propose two methods: Selective Guidance Reweighting (SGR) and Teacher-Inspired Auxiliary Targets (TIAT).
SGR employs two complementary criteria to weigh each sample, enhancing the reliability of teacher trajectory training. In contrast, TIAT provides a regularization signal by aligning the student’s learning trajectory with an additional teacher trajectory that has been fine-tuned on a reliable subset of data.

**Strengths:**

The target scenario explored in this work is novel and underexplored. Addressing dataset distillation under noisy-label conditions is a valuable and timely research direction that could inspire further studies in this emerging area.

**Weaknesses:**

1. Figure 1 is unclear; it does not explicitly illustrate how the proposed TIAT module is integrated into the overall pipeline. The illustration should explicitly depict the TIAT component and annotate how the auxiliary loss, L_{aux}, interacts with the student trajectory during training.

2. The motivation of TIAT remains ambiguous. While it is described as "an auxiliary signal during the student’s inner-loop training" in Section 4.1, the implementation (Eq. 13) combines it with an outer loop loss. It would strengthen the paper to clarify whether TIAT’s role is to improve the teacher model (thereby yielding a better trajectory) or to regularize the student’s optimization.

3. If the primary goal of TIAT is to denoise teacher signals, it might be more appropriately applied during the first-stage teacher training rather than the second-stage student training.

4. The experimental evaluation would be stronger if the authors replaced the SGR component with established noisy-label learning approaches such as [1] and [2], in order to assess whether the proposed SGR truly outperforms existing off-the-shelf methods for learning under noisy labels.

[1] Francesco Di Salvo, Sebastian Doerrich, et al. An Embedding is Worth a Thousand Noisy Labels. TMLR 2025.
[2] Yuyin Zhou, Xianhang Li, et al. L2B: Learning to Bootstrap Robust Models for Combating Label Noise. CVPR 2024.

**Questions:**

1. Can the authors clarify how TIAT is integrated into the overall pipeline?

2. What is the exact role of TIAT? Is it designed to improve the teacher’s trajectory or to regularize the student’s optimization?

---

### Official Review · Reviewer_ws47 · 2025-10-30

**Soundness:** 2
**Presentation:** 2
**Contribution:** 2
**Rating:** 2
**Confidence:** 3

**Summary:**

This paper proposes a novel robust dataset distillation framework under noisy supervision. The authors introduce two components to jointly address noise suppression and knowledge preservation without requiring label correction or clean subsets: elective Guidance Reweighting (SGR), a hybrid mechanism that fuses KNN-based local feature consistency with a forgetting-based prior to weight samples during distillation, and Teacher-Inspired Auxiliary Targets (TIAT), which regularize the student model using signals extracted from the teacher’s trajectory. Experiments on multiple datasets demonstrate that the proposed method shows improvements over recent state-of-the-art distillation methods in the presence of noise.

**Strengths:**

+ This paper designs two simple yet effective modules to address the dataset distillation problem under noisy conditions. The SGR module emphasizes the dynamic consistency of the training data and uses a static prior to ensure that clean signals are prioritized during the distillation process. The TIAT module works by mining a reliable subset and incorporating trajectory-level uncertainty.
+ The paper presents convincing experimental results, demonstrating strong performance under various noise conditions.

**Weaknesses:**

1)	The paper's references include many arXiv articles, some of which have conference versions available, yet the arXiv versions are cited instead. The authors should standardize the citation format for papers that have already been published.

2)	The motivation of the paper is not sufficiently explained. In cases where noise is artificially introduced, existing works have already demonstrated better performance compared to the method presented in this paper. Why is dataset distillation necessary for noisy label learning?

3)	The SGR module simply combines two metrics used for mitigating noisy labels by merging them through weighted summation. The experimental section lacks ablation studies on individual metrics to demonstrate the effectiveness of the chosen scores. Additionally, the TIAT module heavily relies on the SSFT score to determine the reliable subset, and any failure cases in the SSFT score could potentially cause interference. This point requires more discussion.
4)	Experimental results are severely insufficient.

(i) Missing comparisons with recent trajectory-matching distillation methods (e.g., SelMatch[1]).

(ii) No experiments under clean labels (without injected noise) to verify the method does not misidentify clean data as noisy.

(iii) Experiments cover only CIFAR-100N/100/10—why is CIFAR-10N omitted? Table 1 is confusing: are the CIFAR-100N and CIFAR-100 results merged?

(iv) Evaluations are limited to low-resolution datasets; results on medium-resolution (Tiny-ImageNet) and high-resolution (ImageNette, etc.) are missing.

(v) Lacks tests across diverse architectures (e.g., ResNet-10, DenseNet-121).

[1] Lee, Yongmin, and Hye Won Chung. "SelMatch: Effectively Scaling Up Dataset Distillation via Selection-Based Initialization and Partial Updates by Trajectory Matching." Forty-first International Conference on Machine Learning.

[2]Cheng L, Chen K, Li J, et al. Dataset distillers are good label denoisers in the wild[J]. arXiv preprint arXiv:2411.11924, 2024.

**Questions:**

1) Figure clarity. Figure 2 is hard to interpret. Please present these results in a tabular format analogous to Table 1
2) The $\bar{y}_{i}$ in Equation (4) is not introduced earlier. Please check and define it clearly
3) In Fig. 1, the leftmost SSFT label overlaps the connecting line. The “Trajectory Matching” block still has a white background.
4) The paper’s motivation appears to stem from Cheng et al. [2], but there is little explanation in this work. The authors are encouraged to clarify the motivation and provide further experiments.

---

### Official Review · Reviewer_2kZr · 2025-10-31

**Soundness:** 4
**Presentation:** 3
**Contribution:** 3
**Rating:** 6
**Confidence:** 3

**Summary:**

This paper tackles the problem of dataset distillation robustness in the presence of label noise. It proposes a new framework, "Robust Trajectory Distillation," which aims to suppress noise and preserve knowledge without relying on label correction or clean subsets. The framework features two core complementary components: 1) Selective Guidance Reweighting (SGR), which improves teacher signal quality using a hybrid reweighting mechanism that combines global forgetting patterns and local feature consistency; and 2) Teacher-Inspired Auxiliary Targets (TIAT), which enhances student learning by mining a high-confidence subset based on multi-teacher consensus and using it to generate a "cleaner" auxiliary target. Experiments show the method significantly outperforms SOTA distillation methods across various noise settings.

**Strengths:**

1. **Problem Significance and Novelty:** The paper tackles a highly practical and challenging problem: dataset distillation under label noise. This area is relatively underexplored compared to standard noisy label learning. The proposed SGR + TIAT framework is a novel solution that addresses the problem from two complementary angles: improving "teacher signal quality" and enhancing "knowledge transmission".
2. **Strong Empirical Results:** The method consistently and significantly outperforms SOTA baselines (RCIG, DANCE, DATM) across a wide range of noise settings (symmetric, asymmetric, real-world) and images per class (IPC) configurations on both CIFAR-10 and CIFAR-100. The performance gap is particularly evident under high noise ratios, which demonstrates the method's strong robustness.

**Weaknesses:**

1. **Scalability to Larger Datasets:** The experiments are limited to CIFAR-10 and CIFAR-100, which are relatively small-scale datasets. It is unclear whether the method can scale effectively to larger, higher-resolution datasets.
2. **Computational Cost:** While the paper claims the framework is "computationally efficient," the architectural design appears to introduce significant computational overhead. Specifically, the dynamic scoring of the SGR mechanism necessitates computationally expensive KNN calculations in the feature space. Crucially, the authors fail to provide a comparison of either the training time or the computational complexity against the baseline methods. Consequently, the claim of efficiency lacks necessary empirical validation.

**Questions:**

1. Can the authors quantify the additional computational overhead introduced by SGR (especially the KNN calculation) and TIAT?
2. Restricting the experiments solely to small-scale datasets such as CIFAR-10 is inadequate (or is not enough), necessitating the expansion of the evaluation to larger datasets.
3. Regarding the Number of Teachers (P): To what extent does the performance of TIAT depend on the number of teacher trajectories, P? The paper does not seem to provide an ablation on the value of P. Is TIAT still effective if P=1 (i.e., only one teacher trajectory)?

---

### Note · Authors · 2025-11-14

I have read and agree with the venue's withdrawal policy on behalf of myself and my co-authors.